# Banana Lectin from *Musa paradisiaca* Is Mitogenic for Cow and Pig PBMC via IL-2 Pathway and ELF1

Roxane L. Degroote [1], Lucia Korbonits [1], Franziska Stetter [1], Kristina J. H. Kleinwort [1], Marie-Christin Schilloks [1], Barbara Amann [1], Sieglinde Hirmer [1], Stefanie M. Hauck [2] and Cornelia A. Deeg [1,*]

[1]   Department of Veterinary Sciences, Ludwig-Maximilians-Universität Munich, 82152 Martinsried, Germany; r.degroote@lmu.de (R.L.D.); Lucia.Korbonits@tiph.vetmed.uni-muenchen.de (L.K.); franzi.stetter@gmx.net (F.S.); K.Kleinwort@tiph.vetmed.uni-muenchen.de (K.J.H.K.); Marie.Schilloks@tiph.vetmed.uni-muenchen.de (M.-C.S.); baerbl.amann@tiph.vetmed.uni-muenchen.de (B.A.); sieglinde.hirmer@tiph.vetmed.uni-muenchen.de (S.H.)
[2]   Helmholtz Center Munich, German Research Center for Environmental Health, Research Unit Protein Science, 80939 Munich, Germany; hauck@helmholtz-muenchen.de
*   Correspondence: Cornelia.Deeg@lmu.de

**Abstract:** The aim of the study was to gain deeper insights in the potential of polyclonal stimulation of PBMC with banana lectin (BanLec) from *Musa paradisiaca*. BanLec induced a marked proliferative response in cow and pig PBMC, but was strongest in pigs, where it induced an even higher proliferation rate than Concanavalin A. Molecular processes associated with respective responses in porcine PBMC were examined with differential proteome analyses. Discovery proteomic experiments was applied to BanLec stimulated PBMC and cellular and secretome responses were analyzed with label free LC-MS/MS. In PBMC, 3955 proteins were identified. After polyclonal stimulation with BanLec, 459 proteins showed significantly changed abundance in PBMC. In respective PBMC secretomes, 2867 proteins were identified with 231 differentially expressed candidates as reaction to BanLec stimulation. The transcription factor "E74 like ETS transcription factor 1 (ELF1)" was solely enriched in BanLec stimulated PBMC. BanLec induced secretion of several immune regulators, amongst them positive regulators of activated T cell proliferation and Jak-STAT signaling pathway. Top changed immune proteins were CD226, CD27, IFNG, IL18, IL2, CXCL10, LAT, ICOS, IL2RA, LAG3, and CD300C. BanLec stimulates PBMC of cows and pigs polyclonally and induces IL2 pathway and further proinflammatory cytokines. Proteomics data are available via ProteomeXchange with identifier PXD027505.

**Keywords:** PBMC; polyclonal cell stimulation; complement pathway; plant lectin; *Musa paradisiaca*; banana lectin; BanLec; LC-MS/MS; proteomics; ELF1; IL-2

## 1. Introduction

Lectins recognize cell-surface carbohydrates with high specificity and are important for many biological processes [1]. Plant lectins are able to induce various immune responses in vitro and also in vivo [2]. Mitogenic lectins can promote polyclonal stimulation in PBMC [3,4]. Earlier, we showed that a lectin from the banana *Musa paradisiaca* highly stimulated in vitro proliferation of cow PBMC [5]. A subgroup of cows reacted with hyperstimulation to classical polyclonal stimulants like pokeweed mitogen [6,7], concanavalin A [4], but also to BanLec from *M. paradisiaca* [5]. This banana lectin was the first lectin isolated from the large family of Musa [8]. A study on human antibody binding to various foods, unraveled marked binding of IgG4 to banana (*M. paradisiaca*) extract [8]. A mannose-binding lectin was subsequently isolated and named BanLec-I [8]. Bananas are a large family, comprising dessert bananas and plantains [9]. Both belong to the large and perpetually increasing family of *Musaceae* [9,10]. Lectin activity is restricted to certain varieties of banana [11]. These banana lectins belong to the mannose-specific jacalin-related

lectins [12]. Proliferation induced by Musa acuminata (Del Monte banana) lectin led to the expression of the cytokines interferon-$\gamma$, tumor necrosis factor-$\alpha$, and interleukin-2 in mouse splenocytes [11]. BanLec from *M. paradisiaca* stimulated T cell proliferation in man in presence of IL-2 like Concanavalin A [13,14]. The immune pathway activated by BanLec-I was not identified in these studies [13,14]. BanLec also had strong mitogenic activity towards murine T-cells [15]. Upon stimulation, secreted cytokines differed in intensity and quality in association with genetic background of mice [15]. Whereas C57 BL/6-originated splenocytes reacted with interferon gamma (IFNG) production to stimulation with recombinant BanLec, splenocytes of Balb/c mice produced interleukin (IL)-4 and IL-10 [15]. Recently, BanLec is getting more and more attention because it could be shown that it has the ability to inhibit HIV replication [16–18]. This BanLec-HIV recognition is thought to be mediated through mannose epitopes on high-mannose N-glycans [19]. BanLec is one of the most potent anti-viral lectins, but also recognizes bacteria with high-mannose type structures [19,20]. This makes BanLec very interesting as a very potent anti-infective agent with additional immune modulatory capacities [20–23]. Further information about the intervention with the immune system is needed to understand respective processes.

The immune response in large animal models such as pigs and cattle resemble the human immune response more closely than the commonly used laboratory rodent [24]. The shared presence of certain immune cell subtypes as well as similar immunological and pathological processes in several diseases and infections, for instance Tuberculosis and the respiratory syncytial virus (RSV) in cattle or influenza in pigs, are of great value for a better understanding of the immune system in humans [24]. Especially pigs are used as large animal models for several diseases and for basic biomedical research [25–30]. Their advantages are close similarities to humans in terms of size, anatomy, diet and metabolism [27,29–31]. Further, the immune system of pigs resembles that of man for more than 80% of analyzed parameters in contrast to mouse, with only about 10% resemblance [32]. Analyses of the T cell repertoire in pigs revealed that the main differences between porcine and human T cells are the high frequency of both CD4$^+$CD8$\alpha^+$ and TCR-$\gamma\delta$ T cells [33]. Besides these differences, porcine T cells closely resemble human T cells validating the use of pigs for biomedical research into T cells [33].

In previous studies, we could show that in cattle, BanLec from *M. paradisiaca* has a similar stimulatory effect on PBMC as the potent cytokine IL2 [5]. Since the immune system of pigs shows strongest resemblance with humans, and BanLec has potent anti-infective properties, we now aimed at assessing the proliferative response of porcine PBMC after polyclonal in vitro stimulation with BanLec. Moreover, we wanted to get deeper insights into molecular processes accompanying respective immune stimulation.

## 2. Materials and Methods

### 2.1. Animals

In this study, PBMC of 148 healthy dairy cows, from age two to nine, and PBMC of 36 pigs were analyzed. Blood withdrawal was performed according to the German Animal Welfare Act with permission from the responsible authority (Government of Upper Bavaria), following the ARRIVE guidelines and Directive 2010/63/EU. Approval numbers: ROB-55.2-2532.Vet_03-17-106 and ROB-55.2-2532.Vet_02-19-195.

### 2.2. PBMC Preparation from Pigs and Cows

The permission from the dairy farm to use the blood samples from their animals for study purpose was obtained. Blood samples from pigs were obtained either at a local slaughterhouse, or from the chair for molecular animal breeding and biotechnology of Badersfeld. Venous whole blood was collected in sodium-heparin (25.000 I.U.) coated tubes and PBMC were prepared as previously described [5,28]. Briefly, cow blood was diluted 1:2 in PBS (NaCl 136.9 mM, Na$_2$HPO$_4$ $\times$ 2H$_2$O 8.1 mM, KH$_2$PO$_4$ 1.4 mM and KCl 2.6 mM; pH 7.4) and PBMC were isolated by density gradient centrifugation (23 °C, 500$\times$ *g*, 25 min, brake off) using Pancoll separating solution (PanBiotech, Aidenbach, Germany). PBMC

were obtained from intermediate phase, washed twice in PBS, and immediately used for in vitro stimulation. Pig PBMC were isolated by density gradient centrifugation (23 °C, $500 \times g$, 25 min, brake off) of undiluted venous whole blood using Pancoll separating solution (PanBiotech, Aidenbach, Germany). PBMC were obtained from intermediate phase, washed twice in PBS (NaCl 136.9 mM, $Na_2HPO_4 \times 2H_2O$ 8.1 mM, $KH_2PO_4$ 1.4 mM and KCl 2.6 mM; pH 7.4), and immediately used for in vitro stimulation.

### 2.3. Polyclonal Stimulation of Bovine and Porcine PBMC with Mitogens

Optimal mitogen concentration for each species was separately determined beforehand or used as previously published [5,28] To assess the proliferative response to *M. paradisiaca* lectin (BanLec, L1410; Vector, Stuttgart, Germany; pigs with 1 µg/mL, cows with 5 µg/mL) and two control mitogens, eight experiments with a total of 36 pigs (two technical replicates) and 35 independent experiments with a total of 148 cows were performed. In every assay, technical replicates (duplicates or triplicates) were generated for each tested animal. Mean values of technical replicates were then used for further statistical analysis. As positive controls, PBMC were either stimulated by pokeweed mitogen (PWM; Sigma, Taufkirchen, Germany; pigs with 1 µg/mL, cows with 5 µg/mL) or concanavalin A (ConA; Sigma; pigs with 1 µg/mL, cows with 5 µg/mL). After 34 h of stimulation, cells were pulsed for 14 h with 0.05 mCi/well [methyl-3H]-thymidine (Perkin Elmer, Hamburg, Germany), harvested and counts per minute were measured using a Microbeta (Perkin Elmer). Proliferation rate was expressed as the ratio of 3H-thymidine incorporation by stimulated cells with respect to unstimulated cells.

### 2.4. Secretome and Cell Lysate Preparations from Pigs for Differential Proteome Analysis

For differential proteome analyses, porcine cells were resuspended in RPMI 1640 (PanBiotech) with 1% Penicillin-Streptomycin (PanBiotech), but without serum. Pig PBMC ($2 \times 10^7$ cells for proteome analyses) were then stimulated with 1 µg/mL BanLec at 37 °C and 5% CO2 and unstimulated controls were incubated under same conditions. After 48 h, cells were washed twice with PBS, supernatants were discarded and PBMC immediately fractionated before proteomic analysis. Secretome samples were centrifuged ($1000 \times g$) after 48 h of incubation, the supernatants collected and subsequently digested in a filter-aided sample preparation (FASP). For control of proliferation rate of the porcine PBMC, proliferation assays were performed simultaneously. Therefore, PBMC ($1 \times 10^5$ cells/well) were stimulated in triplicates for 48 h with either 1 µg/mL BanLec, ConA or PWM in 96-well plates as described [28]. Cellular lysates of PBMC and secretome samples were digested by a modified FASP protocol as described [34,35]. Briefly, eluates were diluted 1:10 with 0.1 M Tris/HCl pH 8.5 and 100 mM dithiothreitol was added for 30 min at 60 °C. After cooling down, UA buffer (8 M urea and 1 M Tris-HCl pH 8.5 diluted in HPLC-grade water) and 300 mM iodoacetamide were added and incubated for 30 min at room temperature in the dark. Eluates were transferred to 30 kD cut-off centrifuge filters (Sartorius, Göttingen, Germany) and washed five times with UA-buffer and two times with ABC buffer (50 mM $NH_3HCO_3$ diluted in HPLC-grade water). After washing, proteins were subjected to proteolysis for 2 h at room temperature with 0.5 µg Lys C in ABC-buffer followed by addition of 1 µg trypsin and incubation at 37 °C overnight. Peptides were collected by centrifugation and acidified with 0.5% trifluoroacetic acid.

### 2.5. Mass Spectrometric Analysis and Label-Free Quantification

Porcine PBMC and supernatant samples were digested separately by a modified FASP protocol as described [35]. Eluted peptides were analyzed on a Q Exactive HF-X mass spectrometer (Thermo Fisher Scientific, Waltham, MA, USA) in the data-dependent mode. Approximately 0.5 µg peptides per sample were automatically loaded to the online coupled ultra-high-performance liquid chromatography (UHPLC) system (UltiMate 3000 –RSLCnano System, Thermo Fisher Scientific). A nano trap column was used (300-µm ID × 5 mm, packed with Acclaim PepMap100 C18, 5 µm, 100 Å; LC Packings, Sunnyvale,

CA, USA) before separation by reversed phase chromatography (Acquity UHPLC M-Class HSS T3 Column 75 μm ID × 250 mm, 1.8 μm; Waters, Eschborn, Germany) at 40 °C. Peptides were eluted from the column at 250 nL/min using increasing ACN concentration (in 0.1% formic acid) from 3% to 41% over a linear 95-min gradient. MS spectra were recorded at a resolution of 60,000 with an AGC target of $3e^6$ and a maximum injection time of 50 ms from 300 to 1500 m/z. From the MS scan, the 15 most abundant peptide ions were selected for fragmentation via HCD with a normalized collision energy of 28, an isolation window of 1.6 m/z, and a dynamic exclusion of 30 s. MS/MS spectra were recorded at a resolution of 15,000 with a AGC target of $1e^5$ and a maximum injection time of 50 ms. Unassigned charges, and charges of +1 and above +8 were excluded from precursor selection.

Data recorded for lysates and secretomes were processed separately. Proteome Discoverer 2.4 software (Thermo Fisher Scientific, Dreieich, Germany; version 2.4.1.15) was used for peptide and protein identification via a database search (Sequest HT search engine) against Ensembl Pig database (Release 75, Sscrofa10.2; 25,859 sequences), considering full tryptic specificity, allowing for up to two missed tryptic cleavage sites, precursor mass tolerance 10 ppm, fragment mass tolerance 0.02 Da. Carbamidomethylation of Cys was set as a static modification. Dynamic modifications included deamidation of Asn and Gln, oxidation of Met; and a combination of Met loss with acetylation on protein N-terminus. Percolator was used for validating peptide spectrum matches and peptides, accepting only the top-scoring hit for each spectrum, and satisfying the cutoff values for FDR < 1%, and posterior error probability <0.01. The final list of proteins complied with the strict parsimony principle.

The quantification of proteins, after precursor recalibration, was based on abundance values (intensity) for unique peptides. Abundance values were normalized to the total peptide amount to account for sample load errors. The protein abundances were calculated summing the abundance values for admissible peptides and these abundances were used for ratio calculations. Ratios above 100-fold and below 0.1-fold were combined into these bins. The statistical significance of the ratio change was ascertained employing the approach described [36], which is based on the presumption that we look for expression changes for proteins that are just a few in comparison to the number of total proteins being quantified. The quantification variability of the non-changing "background" proteins can be used to infer which proteins change their expression in a statistically significant manner.

### 2.6. Data Analysis

For determination of Gaussian distribution of proliferation data, Kolmogorow–Smirnow (KS) test was used. Since analyzed data showed normal distribution (KS test was significant; $p < 0.05$), paired *t*-test was used for statistical analysis. In pathway enrichment analyses, Bonferroni corrected *p*-values against whole genome as a background were generated. Statistical probabilities were considered significant at ($p \leq 0.05$ = *; $p \leq 0.01$ = **; $p \leq 0.001$ = ***). All proteins identified by mass spectrometric analysis, that showed significant ($p \leq 0.05$) abundance changes after BanLec stimulation were further analyzed. Functional enrichment analysis of cell- and secretome-derived data was performed with open source software FunRich [37], version 3.1.4, available at http://funrich.org/download (accessed on 15 July 2021). *K*-means clustering was visualized with open source software String version 11.0, available at https://string-db.org/ (accessed on 22 July 2021).

### 2.7. Data Availability

The mass spectrometry proteomics data have been deposited to the ProteomeXchange Consortium (http://www.proteomexchange.org/) (accessed on 22 July 2021) via the PRIDE [38] partner repository with the dataset identifier PXD027505.

## 3. Results

### 3.1. BanLec Significantly Stimulates Proliferation of Bovine and Porcine PBMC In Vitro

Co-incubation of bovine PBMC with BanLec for 48 h resulted in a marked proliferative response, with an average proliferation factor of 55.10 (SEM $\pm$ 1.91) (Figure 1A, yellow bar). Strongest proliferation of bovine PBMC was induced by ConA (proliferation factor 64.93 (SEM $\pm$ 2.27), Figure 1A, blue bar), whereas stimulation with PWM resulted in lowest proliferation rate (proliferation factor 37.01 (SEM $\pm$ 1.25), Figure 1A, green bar). Porcine PBMC on the other hand, showed strongest proliferation response to BanLec (proliferation factor 123.80 (SEM $\pm$ 16.20), Figure 1B, yellow bar), followed by ConA (proliferation factor 81.25 (SEM $\pm$ 9.09), Figure 1B, blue bar) and PWM (proliferation factor 57.80 (SEM $\pm$ 5.72), Figure 1B, green bar). Interestingly, proliferative response of porcine PBMC to BanLec was more than twice as high as response of bovine PBMC (Figure 1, yellow bars).

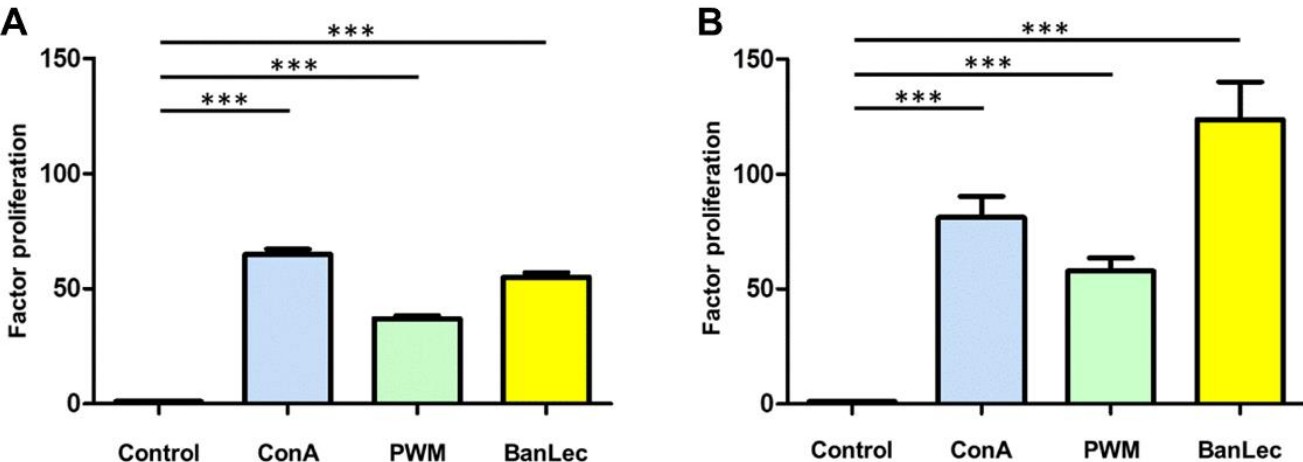

**Figure 1.** PBMC of 148 dairy cows (**A**) and 36 pigs (**B**) showed a significantly higher proliferation rate (*** $p < 0.001$, paired *t*-test) after polyclonal in vitro stimulation with ConA (blue bar), PWM (green bar) and BanLec (yellow bar).

### 3.2. Significantly Changed Proteome in BanLec Stimulated Porcine PBMC

Since porcine PBMC showed such a strong proliferative response to BanLec, we were interested in proteome changes induced by BanLec stimulation and their biological relevance. From a total of 3955 proteins identified in porcine PBMC, 459 showed a significantly differential expression (Table S1). Functional enrichment analysis of these differentially abundant proteins revealed the impact of BanLec associated proteome changes on several biological processes and pathways as well as molecular function, protein domains and transcription factors (Table S1): Proteins associated to the biological processes 'protein metabolism' (54 mapped proteins) and 'metabolism' (64 mapped proteins) were significantly enriched in the proteome of BanLec stimulated porcine PBMC (Table 1 and Table S1). Furthermore, we identified 17 significantly enriched biological pathways, among which "cell cycle phases" and "DNA synthesis" were overrepresented (Table S1). Cysteine-type peptidase activity was the only significantly enriched molecular function associated to proteins with changed abundance after BanLec stimulation (Table S1). Additionally, protein domains Pept_C1, MCM and DUF4217 were enhanced (Table S1). Interestingly, one transcription factor, namely "E74 like ETS transcription factor 1 (ELF1)", was significantly enriched. From all differentially abundant proteins associated to BanLec stimulation, 31 candidates were associated to ELF1 (Table 2 and Table S1).

**Table 1.** Significantly enriched biological processes in BanLec stimulated porcine PBMC.

| Biological Process | No. of Genes in the Dataset | Percentage of Genes | Fold Enrichment | Bonferroni Method | Proteins Mapped from Input Data Set (Porcine PBMC Proteome) |
|---|---|---|---|---|---|
| Protein metabolism | 54 | 12.3 | 1.7 | 0.02 | CBLL1; CLUH; EIF4EBP1; EIF4ENIF1; F8A1; FBXO7; FBXW11; GRSF1; MMP14; MRPL2; MRPL24; MRPL47; MRPL54; MRPS18B; MRPS21; MRPS26; PSMD5; TARS2; TMUB1; UBE2C; USP8; CST7; DTX3L; LGMN; PRTN3; SERPINB2; AGBL3; CUL3; DPEP2; HERC6; ITIH2; MRPL28; RPS15A; ZNRF2; ST6GALNAC2; CTSZ; CTSH; ASNS; HARS2; RPL38; NPM3; ASF1B; ELANE; SELENBP1; CTSA; CTSC; PPIF; CTSB; CTSS; SERPINB10; CELA3A; CSTB; PCNP; MRPS9 |
| Metabolism | 64 | 14.5 | 1.6 | 0.04 | BCKDHB; COX6A1; CYP51A1; DDX19B; DHCR24; EBP; FAR1; GALT; HCCS; LPGAT1; MAN1A1; MAN2A1; MMAB; NAA40; NUDT15; OXA1L; PDK3; PLCB3; PRMT3; PSAT1; PYCR1; QTRT1; RFK; SOAT1; ACAD9; ADA; ARSA; FUCA2; GAA; GGH; NPL; CHST11; CLPB; DHCR7; PMVK; NAAA; TYMS; IFI30; ACSL4; F13A1; FUCA1; ALOX15; AK4; PAPSS2; MSRA; HEXB; NPC2; SCD; FASN; CA2; MAN2A2; HMGCS1; ASAH1; MTHFD2; CA1; PPT1; PRPSAP1; GALM; HEXA; GALNS; SMPDL3A; ACAD8; TGM3; NAGA |

**Table 2.** Significantly enriched transcription factor in BanLec stimulated porcine PBMC.

| Transcription Factor | No. of Genes in the Dataset | Percentage of Genes | Fold Enrichment | Bonferroni Method | Proteins Mapped from Input Data Set (Porcine PBMC Proteome) |
|---|---|---|---|---|---|
| ELF1 | 31 | 7.7 | 2.0 | 0.04 | ASAP2; BRI3BP; CDV3; EIF4EBP1; EIF4ENIF1; KLF13; LAG3; LIMS1; MRPL24; NUFIP2; ORC4; RAC1; RGS3; RSRC2; SHOC2; STX5; TBC1D2B; TXNIP; WDHD1; ZMYND8; SERPINB2; SYNE1; BMI1; RCL1; THADA; FCGR2B; ACSL4; UBAP2L; CTSA; HNMT; APOBR |

*3.3. Functionally Enriched Proteins in Secretome of BanLec Stimulated PBMC*

After polyclonal stimulation with BanLec, we also found significant ($p < 0.05$) changes in PBMC secretome. From a total of 2857 proteins identified in secretome of porcine PBMC, 231 showed significant abundance changes after BanLec stimulation (Table S2). Functional enrichment analyses of these candidates revealed that the biological process 'immune response' was the top and only significantly enriched biological process in the secretome of BanLec stimulated pig PBMC, with 21 mapped proteins (Table 3 and Table S2).

**Table 3.** Significantly enriched biological process in secretome of BanLec stimulated porcine PBMC.

| Biological Process | No. of Genes in the Dataset | Percentage of Genes | Fold Enrichment | Bonferroni Method | Proteins Mapped from Input Data Set (Porcine PBMC Secretome) |
|---|---|---|---|---|---|
| Immune response | 21 | 9.7 | 3.0 | 0.001 | CD226; CD27; IFNG; IL18; IL2; CXCL10; LAT; ICOS; IL2RA; LAG3; CD300C; CD69; C1QB; C1QA; CSF3; AOAH; CFD; C4BPA; CFP; C3; IL1B |

### 3.4. Immune Response Proteins Enriched in BanLec Stimulated PBMC Secretome Cluster to Complement Activation and Jak-STAT Signaling Pathway

*K*-means clustering of immune response related, differentially regulated proteins in BanLec secretome resulted in two clusters (Figure 2). One cluster (Figure 2, red dots) comprised proteins related to the functions of cytokine–cytokine receptor interaction, Jak-STAT signaling pathway and positive regulation of activated T cell proliferation. The second cluster (Figure 2, green dots) included several members of complement cascade. For the top ten enriched proteins associated to the biological process "immune response", relations and known interactions were analyzed with String protein software (version 11, species sus scrofa). For nine of these proteins, functional relations were found, which were especially strong for IL2, IL2RA, IFNG, IL18, CD27, and CXCL10 (Figure 3).

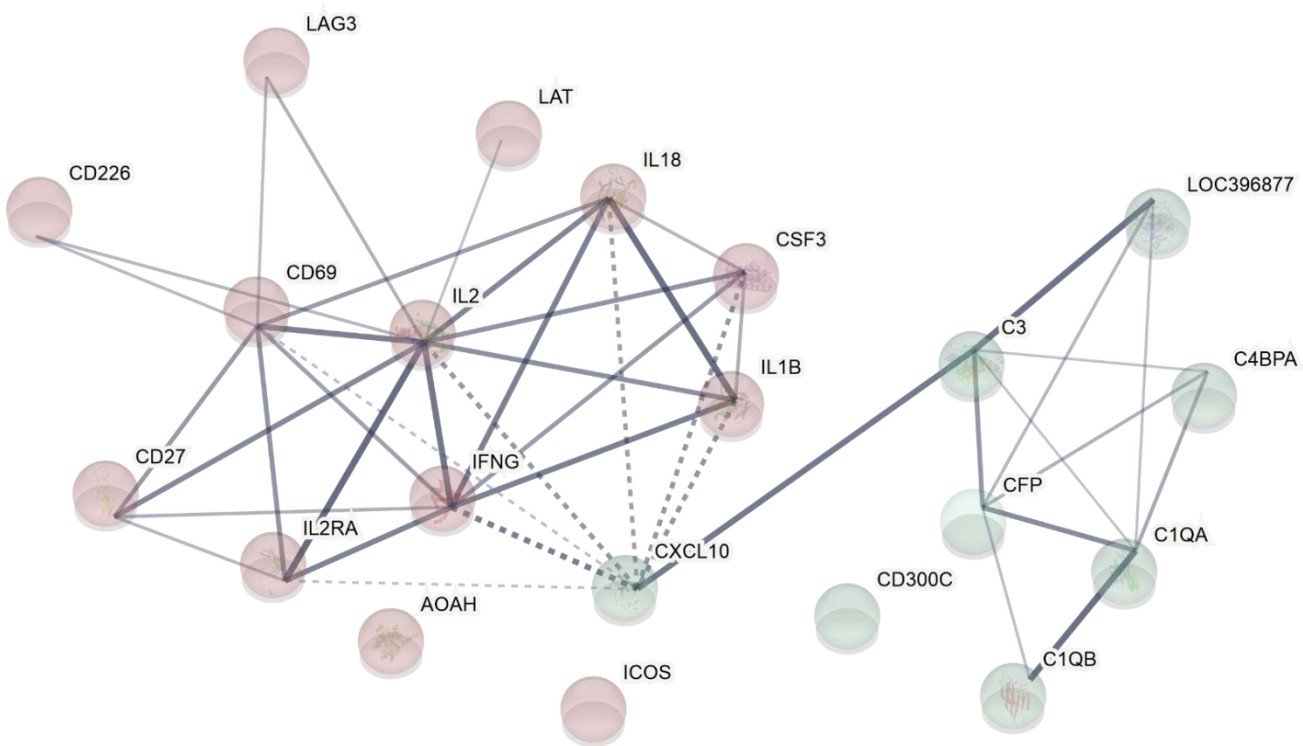

**Figure 2.** Enriched immune response candidates from PBMC secretome were used for *k*-means clustering, resulting in two distinct clusters (red and green), visualized with open source software String version 11.0.

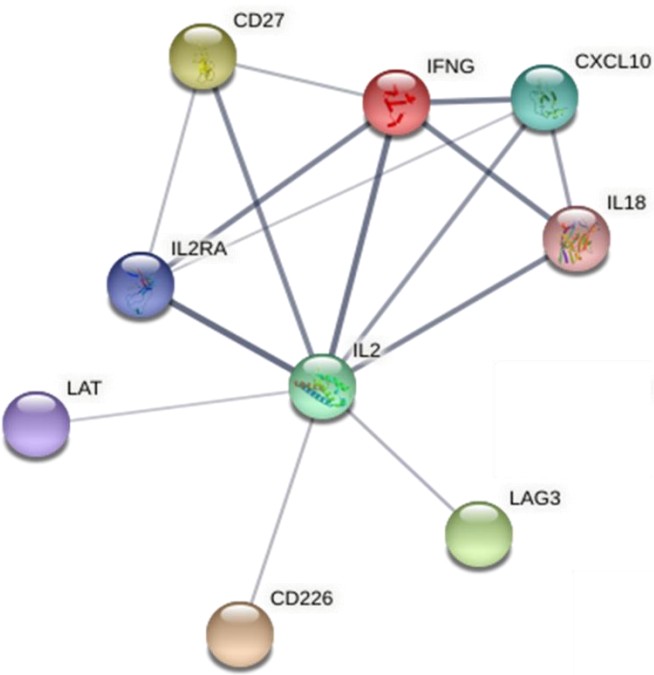

**Figure 3.** Interaction of top ten enriched immune proteins in BanLec stimulated porcine PBMC secretome, analyzed by open source software String version 11.0.

## 4. Discussion

BanLec is able to induce a marked in vitro proliferation in bovine and porcine PBMC. We observed a difference in overall proliferation capacity between cows and pigs in this study, which was greatest for BanLec: this mitogen caused a more than two-fold higher proliferation rate in pig PBMC compared to PBMC from cows. Moreover, in porcine PBMC, BanLec induced the strongest reaction of all polyclonal activators tested. Therefore, we were interested in the immune pathways which are triggered through BanLec in porcine PBMC. With differential proteome analyses, we detected several proteins with increased abundance after BanLec stimulation, which, amongst others, associated to the biological processes protein metabolism and metabolism, indicating a high rate of reproduction in proliferating PBMC. Analyses of transcription factors surfaced a single factor, ELF1, among the differentially regulated candidate proteins. ELF1 acts as regulator of hematopoiesis and energy metabolism in chromatin immunoprecipitation assays and murine liver cells [39]. Further, ELF1 triggers the NF-κB pathway activation involved in human T leukemic cells [40] and is involved in the regulation of several T- and B-cell-specific genes, through the transcriptional activators of BLK and SRC kinases [41]. In pigs, ELF1 was shown to be associated with growth processes [42]. In our study, we could show that BanLec activates ELF1 in porcine PBMC. Although we cannot exactly specify whether this BanLec associated activation of ELF1 has regulatory impact on BLK and SRC kinases from pig PBMC, we did detect BLK and several members of the SRC kinase family in our dataset. Part of these candidates did not change their abundance after BanLec stimulation (BLK, SYK, CSK, LYN). Others showed significant changes, such as the SRC kinase Feline Gardner-Rasheed sarcoma viral oncogene homolog (FGR), which was significantly lower abundant in BanLec stimulated porcine PBMC. FGR protein localizes to plasma membrane ruffles, and functions as a negative regulator of cell migration and adhesion triggered by the beta-2 integrin signal transduction pathway in man [43]. The exact function in the immune response of pigs has not been described so far.

In the secretome of BanLec stimulated porcine PBMC, on the other hand, we detected significantly higher abundant members of SRC family kinases, namely Lymphocyte-specific protein tyrosine kinase (LCK) and FYN oncogene related to SRC, FGR, YES (FYN). An increase of LCK, a key signaling molecule in the selection and maturation of developing

CD4$^+$ T-cells [44,45] was already shown by genome sequencing of porcine PBMC stimulated with lipopolysaccharide (LPS) [46]. In our study, LCK was significantly increased (3.2 fold) in the secretome of BanLec stimulated PBMC. A similar increase in the BanLec activated pig PBMC secretome was also observed for FYN (3.3 fold). The function of FYN, an interactor of LYN and inductor of T-cell differentiation and proliferation following T-cell receptor (TCR) stimulation [45], is unknown in pigs so far. In man, FYN inhibits differentiation to Th17 cells [47]. In our opinion, the role of LCK, FYN and ELF1 in pig PBMC deserves further in-depth investigations in the future to clarify their interactions and detailed functions.

The top ten enriched immune proteins present in BanLec stimulated PBMC secretome consisted of CD226, CD27, IFNG, IL18, IL2, CXCL10, LAT, ICOS, IL2RA and LAG3.

CD226, a member of the immunoglobulin superfamily, is an adhesion molecule that was shown to promote migration, activation, proliferation, differentiation, and function of CD8$^+$ T cells of mice and man [48,49]. But CD226 is widely expressed on various immune cells and therefore plays an important functional role in the immune system [48]. CD226, as a costimulatory factor, plays an important role in the development of various diseases and is currently under thorough investigation for increased understanding of the molecule's clinical relevance [48–50]. The exact distribution and function of CD226 in porcine immune cells was not described so far.

CD27 is a well characterized molecule in pig immune cells [51,52]. It is expressed by all naïve CD8$\alpha^-$ T helper cells but divides CD8$\alpha^+$ T helper cells into a CD27$^+$ and a CD27$^-$ subset [52]. Analyses with polyclonally stimulated PBMC showed that CD8$\alpha^+$CD27$^+$ T helper cells have an intermediate proliferation rate similar to naïve CD8$\alpha^-$CD27$^+$ T helper cells as well as intermediate levels of cytokine production [52]. Therefore, this subpopulation of CD8$\alpha^+$CD27$^+$ T helper cells displayed a phenotype and functional properties of central memory cell like found in humans [52].

IFNG, IL18 and IL2 are proinflammatory cytokines, produced by activated porcine T-cells [52–55]. IL18 is a member of IL-1 cytokine family and plays a protective role in many virus infections [55]. It augments IFNG production in T-cells. IFNG activates effector immune cells and enhances antigen presentation [56]. IL2 is important for T cell proliferation in pigs [54]. In addition to the IL2 cytokine itself, the interleukin 2 receptor alpha (IL2RA) was elevated in our study. IL2RA and IL2RB chains, together with the common gamma chain IL2RG, constitute the high-affinity IL2 receptor. A targeted disruption of the X-linked interleukin-2 receptor gamma chain gene in pigs enables the generation of immune deficient pigs to study the impact of IL2RG for the pig immune response [57]. In this study, only heterozygous pigs survived birth [57]. Il2rg−/Y males had undetectable thymi and reduced peripheral blood T cells [57]. Another study confirmed the severely impaired immune phenotype of Il2rg−/Y pigs with lymphopenia, lymphoid organ atrophy, poor immunoglobulin function, and T- and NK-cell deficiency, underscoring the importance of the IL2 pathway for the immune response [58].

Chemokine (C-X-C motif) ligand 10 (CXCL10) was significantly upregulated in the PBMC secretome as a response to BanLec stimulation. In porcine PBMC, CXCL10 is a proinflammatory cytokine that is involved in a wide variety of processes such as chemotaxis, differentiation and activation of peripheral immune cell [59]. CXCL10 increased in PRRSV infected pig lungs or after stimulation with poly (I:C) [59].

The linker for activation of T cells (LAT) was shown to be required for TCR-mediated signaling. LAT interacts in a feedback loop with Zap70 and Src-family kinases [60]. Src-family kinases, e.g., LCK bind phosphorylated LAT [60]. Since we also detected LCK as being upregulated in BanLec-activated PBMC, this points to activation of this axis in pig PBMC.

With the inducible T-cell co-stimulator ICOS as higher abundant immune response protein in BanLec stimulated PBMC secretome, we identified an important costimulatory molecule for T-cell proliferation and cytokine secretion [61].

Lymphocyte activation gene 3 (LAG3) belongs to the Ig superfamily and is an inhibitory receptor on antigen-activated T-cells [62]. Following TCR engagement, LAG3

associates with CD3-TCR in the immunological synapse and directly inhibits T-cell activation [63]. It also mediates immune tolerance through constitutive expression on a subset of regulatory T-cells and contribution to their suppressive function [63]. Its upregulation in our secretome dataset may point to activation of homeostatic mechanisms in activated pig lymphocytes in order to terminate the inflammatory response through LAG3. Neither LAT, nor ICOS or LAG3 functions in PBMC of pigs are understood to date, therefore, further experiments are needed to clarify their respective roles in the porcine immune response.

## 5. Conclusions

Lectin from *M. paradisiaca* (BanLec) is highly mitogenic for bovine and porcine PBMC, however, compared to bovine cells, pig PBMC react much stronger to BanLec stimulation. With differential proteome analyses, we detected regulated proteins in the pig PBMC proteome and secretome as a response to activation by BanLec. Several immune-related proteins were detected with significantly changed abundance in BanLec stimulated PBMC, pointing to a role of the IL2-pathway, CD226, CD27, LAT, ICOS and LAG3. Further, our study points to an important role of the transcription factor ELF1 in BanLec mediated polyclonal stimulation of pig PBMC. Due to strong similarities of the human and porcine immune system, the pig is an important model organism with high translational value. BanLec is a very interesting lectin, because it has powerful antiviral properties. These properties may be useful for future applications as a possible anti-infective agent. However, BanLec also has the capacity to overstimulate the immune system, resulting in harmful side effects that could make it useless as a drug. These effects need to be assessed in more detail in future studies.

**Supplementary Materials:** The following are available online at https://www.mdpi.com/article/10.3390/immuno1030018/s1, Table S1: porcine PBMC proteome identifications and FunRich analyses of proteins from cellular dataset, Table S2: porcine PBMC seceretome identifications and FunRich analyses of proteins from secretome dataset.

**Author Contributions:** Conceptualization, C.A.D.; methodology, C.A.D., K.J.H.K. and S.M.H.; formal analysis, R.L.D., L.K., F.S., K.J.H.K., M.-C.S., B.A., S.H., S.M.H.; investigation, R.L.D., S.M.H. and C.A.D.; writing—original draft preparation C.A.D.; writing—review and editing, R.L.D. and S.M.H.; visualization, B.A., C.A.D.; supervision, C.A.D. All authors have read and agreed to the published version of the manuscript.

**Funding:** This research received no external funding.

**Institutional Review Board Statement:** The study was approved by the Ethics Committee local authority Regierung von Oberbayern, Munich, permit nos. ROB-55.2-2532.Vet_03-17-106 and ROB-55.2-2532.Vet_02-19-195.

**Data Availability Statement:** The mass spectrometry proteomics data have been deposited to the ProteomeXchange Consortium (http://www.proteomexchange.org/) via the PRIDE [38] partner repository with the dataset identifier PXD027505 (accessed on submission date 22 July 2021).

**Conflicts of Interest:** The authors declare no conflict of interest.

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
