# Peer review of "Banana Lectin from Musa paradisiaca Is Mitogenic for Cow and Pig PBMC via IL-2 Pathway and ELF1"

_2673-5601, doi:10.3390/immuno1030018_

Round 1

Reviewer 1 Report

The manuscript was generally well written. The study was well designed and the results were presented clearly. The discussion is also exhaustive. Below please find minor comments/suggestions to improve the quality of the paper:
Abstract:
Line 16:
Please write the Latin name 'Musa paradisiaca' in italics, this remark applies to the whole text of manuscript.
Introduction:
Line 36: Please remove the dot before ref. 2.
Line 44: please write the Latin name 'Musaceae' in italics.
Please complete the information provided between lines 61-68 with data on studies on cattle.
Materials and Methods
Line 98: Please explain on what basis the Musa paradisiaca lectin doses for pigs and cows were determined; why was it a different concentration? 
Results
Lines 196; 204-205:
According to the figs 1 and 2, the presented values should be presented with a dot, not a comma.
Lines 207-209: The caption under fig. 2 should not be written in italics, if so, please harmonize it with fig. 1. 
Discussion
Line 327: Please remove the dot after 'porcine PBMC'.
Line 343: Two similar words are used in the sentence like 'activation' and 'activated', try to look up a synonym for one of them.

Author Response

we thank the reviewer for their comments. please see attachment for our point to point response.

Reviewer 2 Report

This 5-fold increase in BanLec concentration is already characterized or was arbitrary, and how valid is said intervention <line 103>

The link or the http://funrich.org/index.html program does not work. It would be prudent to mention the date of use of the resource, because it is "out of service or under maintenance" <line 176>

Verify that the data are normal when the value of p in KS is significant <line 179>

There are updated databases that could support the visualization and analysis of interactomes (Reactome for example), in addition to specifying in the methods what were the characteristics of the genes analyzed in String, that is, the top 10 or only IL2 were analyzed as molecule of interest <line 183>

The term << daily tested >> is correct, since it gives the idea that bovines and pigs were monitored, from which samples were taken <line 194>

There is no difference between ConA and BanLec or with PWM, in addition to mentioning the tests performed for the analysis in the figure footer1 <line 197>

Check if the image shown is in pigs or cows. I suggest making a single plot of cattle and pigs, where the great differences between them are shown, the ability of PBMC to proliferate in pigs, which can be observed in plots greater than 100 and in the case of cows close to 60 at the highest level, which shows an "advantage" of this study <line 207>

These data can be graphed, part of the information they contain in the supplementary material, the genes can be added supplementary material and all the information graphed, leaving an illustrative scheme of their findings <tables>

This section must be mentioned in methods, I recommend adding the software <line 253> in the figure caption

This part of the discussion could be better observed in the plots, comparing between them <line 261>

It was random to take these mitogen concentrations between species (pig and cow) increasing the concentration 1: 5 <line 263>

A comparative analysis is not shown that demonstrates this discussion section, since the table is only shown in pigs, a table of the genes in cows is needed in order to differentiate between the two, also between the different inductions with mitogens, that is say between BanLec, ConvA and PWM. <line 273-277>

Only find the RSRC2. The data for the next result are not shown "LCK was significantly increased (3.2 fold) in BanLec stimulated PBMC." <Line 253, 288> Unify between 3.2 and 3.3 fold, also in the tables it is handled with "." or "," for decimal places

All these over-dysregulations can be seen on a volcano plot, data not shown CXCL10, LCK <line 326, 333>

How abundant is ICOS? <line 335>

A Venn diagram or a graph could be made showing the similarities and differential proteins/genes between the interventions performed and the "control", as well as the differences or similarities between pigs and cows. They have many results that can be included with data visualization that would help with understanding complex paths like the ones they mention within the discussion.

There are data described in the manuscript that are not displayed in it or in the supplementary data tables. In the case of the aforementioned Folds, it would be worth mentioning or making a volcano plot that indicates significant differential proteins.

His conclusion is too interesting, however, in the discussion, it is necessary to mention precisely this last point, in addition to mentioning what are the benefits and for whom, as well as the perspectives of the study. Discuss future applications of the study findings. The discussion focuses only on pigs, they left aside the response of the cows, their objective is to evaluate the response of the stimulation in both PBMC induced with BanLec. Furthermore, the differential proteome (volcano plots) between BanLec and its control could be made and explained graphically. In the discussion, add a paragraph that mentions the main differences found in the stimulation between cows and pigs.

Author Response

(The authors gave the same response as above.)

Reviewer 3 Report

Degroote et al. present a quality and well-written experimental manuscript showing that banana lectin from Musa paradisiaca is mitogenic for cow and pig PBMC via IL-2 pathway and ELF1.

Authors aimed to gain deeper insights in the potential for polyclonal stimulation of PBMC from cows and pigs with banana lectin (BanLec) from Musa paradisiaca. They observed that BanLec induced a marked proliferative response in cow and pig PBMC. The response in pigs was even higher than to Concanavalin A. Molecular processes associated with respective responses were examined with differential proteome analyses. Discovery proteomic experiments was applied to BanLec stimulated PBMC and cellular and secretome responses were analyzed with label free LC-MS/MS.

Authors identified 3955 proteins in PBMC. After polyclonal stimulation with BanLec, 459 proteins showed significantly changed abundance in PBMC. In respective PBMC secretomes, 2867 proteins were identified with 231 differentially expressed candidates as reaction to BanLec stimulation.

Authors found that BanLec significantly stimulates proliferation of bovine PBMC in vitro. Porcine PBMC proliferate stronger to BanLec stimulation than to ConA and Significantly changed proteome in BanLec stimulated porcine PBMC. Functionally enriched proteins in secretome of BanLec stimulated PBMC. Immune response proteins enriched in BanLec stimulated PBMC cluster to complement activation and Jak-STAT signaling pathway

Finally, authors conclude that BanLec stimulates PBMC of cows and pigs polyclonally and induces IL2 pathway and further proinflammatory cytokines.

Overall, the manuscript is valuable for the scientific community and should be accepted for publication.

Other comments:

1) Please check for minor typos throughout the manuscript.

Author Response

Thank you for appreciating our study. As suggested by this reviewer, we checked for typos throughout the manuscript and corrected these accordingly. All changes made to the manuscript are displayed in blue font.